# Estrogen signaling in arcuate *Kiss1* neurons suppresses a sex-dependent female circuit promoting dense strong bones

Candice B. Herber[1], William C. Krause[1], Liping Wang[2], James R. Bayrer[3], Alfred Li[2], Matthew Schmitz[4], Aaron Fields [5], Breanna Ford [6], Zhi Zhang[7], Michelle S. Reid[7], Daniel K. Nomura[6], Robert A. Nissenson[2], Stephanie M. Correa[1,7] & Holly A. Ingraham[1]

Central estrogen signaling coordinates energy expenditure, reproduction, and in concert with peripheral estrogen impacts skeletal homeostasis in females. Here, we ablate estrogen receptor alpha (ERα) in the medial basal hypothalamus and find a robust bone phenotype only in female mice that results in exceptionally strong trabecular and cortical bones, whose density surpasses other reported mouse models. Stereotaxic guided deletion of ERα in the arcuate nucleus increases bone mass in intact and ovariectomized females, confirming the central role of estrogen signaling in this sex-dependent bone phenotype. Loss of ERα in *kisspeptin* (*Kiss1*)-expressing cells is sufficient to recapitulate the bone phenotype, identifying *Kiss1* neurons as a critical node in this powerful neuroskeletal circuit. We propose that this newly-identified female brain-to-bone pathway exists as a homeostatic regulator diverting calcium and energy stores from bone building when energetic demands are high. Our work reveals a previously unknown target for treatment of age-related bone disease.

[1] Department of Cellular and Molecular Pharmacology, University of California San Francisco, San Francisco, CA 94158, USA. [2] VA Medical Center Endocrine Unit and Bone Imaging Core Facility, University of California San Francisco, San Francisco, CA 94158, USA. [3] Department of Pediatrics, University of California San Francisco, San Francisco, CA 94158, USA. [4] Graduate Program in Developmental Biology, School of Medicine University of California San Francisco, San Francisco, CA 94158, USA. [5] Department of Orthopedic Surgery, School of Medicine Mission Bay Campus University of California San Francisco, San Francisco, CA 94158, USA. [6] Department of Nutritional Sciences and Toxicology, University of California, Berkeley Berkeley, CA 94720, USA. [7] Department of Integrative Biology and Physiology, University of California Los Angeles, Los Angeles, CA 90095, USA. These authors contributed equally: Candice B. Herber, William C. Krause. Correspondence and requests for materials should be addressed to S.M.C. (email: stephaniecorrea@ucla.edu) or to H.A.I. (email: holly.ingraham@ucsf.edu)

The sex steroid hormone estrogen is critical for balancing energy allocation and expenditure to ensure maximal reproductive fitness. Peripheral estrogen is also an important regulator of skeletal homeostasis. In male and female rodents, circulating 17β-estradiol (E2) actively stimulates cancellous bone formation through estrogen receptor alpha (ERα)[1,2]. Indeed, chronic administration of E2 to intact or ovariectomized (OVX) females can lead to significant increases in trabecular bone mass[3,4]. On the other hand, central estrogen signaling appears to negatively impact female bone metabolism as suggested by the modest increase in trabecular bone mass following central loss of ERα using the brain-specific, but problematic Nestin-Cre[5,6]. Trabecular and cortical bone volume are also modestly elevated after deleting ERα using the non-inducible and developmentally promiscuous POMC-Cre[7,8]. In both of these mouse models (Esr1[Nestin-Cre] and Esr1[POMC-Cre]), the elevated bone mass in females vanishes following ovariectomy, underscoring the essential role of gonadal sex-steroids in generating and/or maintaining these bone phenotypes. Independent of estrogen signaling, manipulating NPY or AgRP ARC neurons also modestly influences bone metabolism, at least in male mice[9,10].

Within the medial basal hypothalamus (MBH) ERα is highly enriched in two anatomically and functionally distinct neuronal clusters, the arcuate nucleus (ARC) and the ventral lateral region of the ventromedial hypothalamus (VMHvl). Estrogen signaling in the female MBH promotes a catabolic energy state by regulating distinct aspects of energy balance[11–13]. Indeed, conditional mouse models in which ERα is deleted in some, but not all, ARC or VMHvl neurons suggest that these two estrogen-responsive brain modules separately regulate energy balance[11,12]. Partial loss or pharmacological blockade of ERα in the VMH lowers energy expenditure by decreasing brown adipose tissue (BAT) thermogenesis[12,14] and lowering physical activity[11]. Within the ARC, ERα is expressed in multiple cell types, each expressing a signature neuropeptide or neurotransmitter system. Estrogen signaling in the POMC lineage[15], is thought to limit food intake as inferred from deletion of ERα using the POMC-Cre (Esr1[POMC-Cre])[12]. Most kisspeptin (Kiss1) neurons in the ARC also express ERα. Kisspeptin itself regulates puberty and fertility in both male and female mice[16,17], as well as in humans[18]. However, ERα signaling dynamically regulates Kiss1 ARC neurons by silencing Kiss1 expression and restraining the onset of female, but not male pubertal development[19]. Deleting ERα using different Kiss1-Cre alleles upregulates Kiss1[20], accelerates pubertal onset in female mice[19], and increases both inhibitory and excitatory firing of Kiss1 ARC neurons[21,22]. While ERα is absent in most but not all NPY/AgRP neurons[23,24], these nutritional sensing neurons project to and inhibit Kiss1 ARC neurons[25].

Given the cellular complexity of estrogen-responsive neurons in the ARC (and VMHvl), we leveraged the Esr1[Nkx2-1Cre] mouse model, in which all ERα in the MBH is eliminated via Nkx2-1-driven Cre recombinase[11] to ascertain how ERα in the MBH influences energy expenditure, food intake, reproduction and possibly skeletal homeostasis. We then confirmed that any observed phenotypes were neural in origin using stereotaxic delivery of AAV2-Cre to acutely ablate ERα in either the ARC or VMHvl in adult female mice. We found that eliminating ERα in the ARC resulted in a robust and sex-dependent high mass bone phenotype. Importantly, we go on to define Kiss1 neurons as the critical estrogen-responsive subpopulation in the ARC for promoting this remarkable, robust bone mass in female mice.

## Results

### Altered energetic homeostasis in female Esr1[Nkx2-1Cre] mice. In Esr1[Nkx2-1Cre] mice, ERα is efficiently eliminated in the entire MBH in both male and female brains, including the ARC and VMHvl, but is largely maintained in the anteroventral periventricular nucleus (AVPV), preoptic area, nucleus tractus solitarius, and medial amygdala (Fig. 1a and Supplementary Figure 1A). Peripheral tissues including the lung, thyroid, BAT, and pituitary maintained Esr1 expression in mutant females compared to wild-type females (Supplementary Figure 1B). Deleting ERα in the MBH depleted primordial follicles, and led to female infertility, and uterine imbibition (Supplementary Figures 1C–E). Mutant females displayed a small but significant increase in body weight, which was less robust than reported for Esr1[POMC-Cre12], whereas body weights for mutant males decreased (Fig. 1b). Food intake was unchanged in both sexes (Fig. 1c).

Esr1[Nkx2-1Cre] females exhibited a sex-dependent change in energy balance that was entirely absent in male mice. The lean mass of mutant females was significantly higher than control floxed (Esr1[fl/fl]) littermates (Fig. 1d) and was accompanied by decreased physical activity during the dark phase (Fig. 1e and Supplementary Figure 2C). Although highly significant, increased lean mass observed in mutant females failed to change their overall mobility and muscle strength as measured by rotarod and grip strength assays, respectively (Supplementary Figure 2D, E). Blunted BAT thermogenesis was observed in mutant females as evidenced by whitening of BAT and decreased Ucp1 levels; circulating catecholamines were not lower (Fig. 1f and Supplementary Figures 2F, G). Serum leptin levels were also unchanged (Fig. 1g). Thus, these data reveal that central estrogen signaling in this brain region promotes a sex-dependent negative energy state in females in the absence of any change in feeding behavior. This unexpected finding implies that the hyperphagia reported for Esr1[POMC-Cre] mice might result from selective or ectopic activity of POMC-Cre in non-ARC neurons[8].

### Elevated bone density in female Esr1[Nkx2-1Cre] mice. Strikingly, bone mineral density (BMD), as determined by dual X-ray absorptiometry (DEXA), was significantly elevated in Esr1[Nkx2-1Cre] females, but not males (Fig. 1h), consistent with the sex-dependent significant increases in lean mass. Further analyses of femoral bone, using three-dimensional high resolution micro-computed tomography (μCT), confirmed a striking increase in trabecular bone mass and microarchitecture in older Esr1[Nkx2-1Cre] females compared to control littermates (Fig. 2a). Mutant females exhibited a ~500% increase in fractional bone volume in the distal femur, rising from 11 to 52 bone volume/tissue volume (BV/TV) (%) (Fig. 2a). A similar trend was found for vertebral bone (Supplementary Figure 3A). Accompanying structural changes included increases in trabecular number and thickness and reduced trabecular separation (Fig. 2a). Mutant females also exhibited a significant increase in cortical thickness but a modest decrease in tibial and femoral length (Supplementary Figure 3B). This striking skeletal phenotype is sex-dependent, as no changes in bone mass were observed in Esr1[Nkx2-1Cre] males (Fig. 2b). Further, unlike the 20% increase in femoral bone mass reported for Esr1[POMC-Cre] and Esr1[Nestin-Cre] mice that vanishes in OVX females, bone parameters in Esr1[Nkx2-1Cre] females remained elevated 5 weeks following ovariectomy (Fig. 2c). In fact, no significant changes in serum sex steroids (E2, T) were detected in 4–5-week-old mutant females when the high bone mass phenotype is clearly present (Fig. 2d, f). The percentage of bone loss in OVX female mutants is higher than their wild-type littermates (72% versus 43%), consistent with the finding by others that a higher starting %BV/TV at baseline results in higher OVX-mediated bone loss[26]. Pituitary and thyroid hormones in mutant females were also unchanged at 7–8 weeks of age (Supplementary Figure 3C). Removing circulating androgens

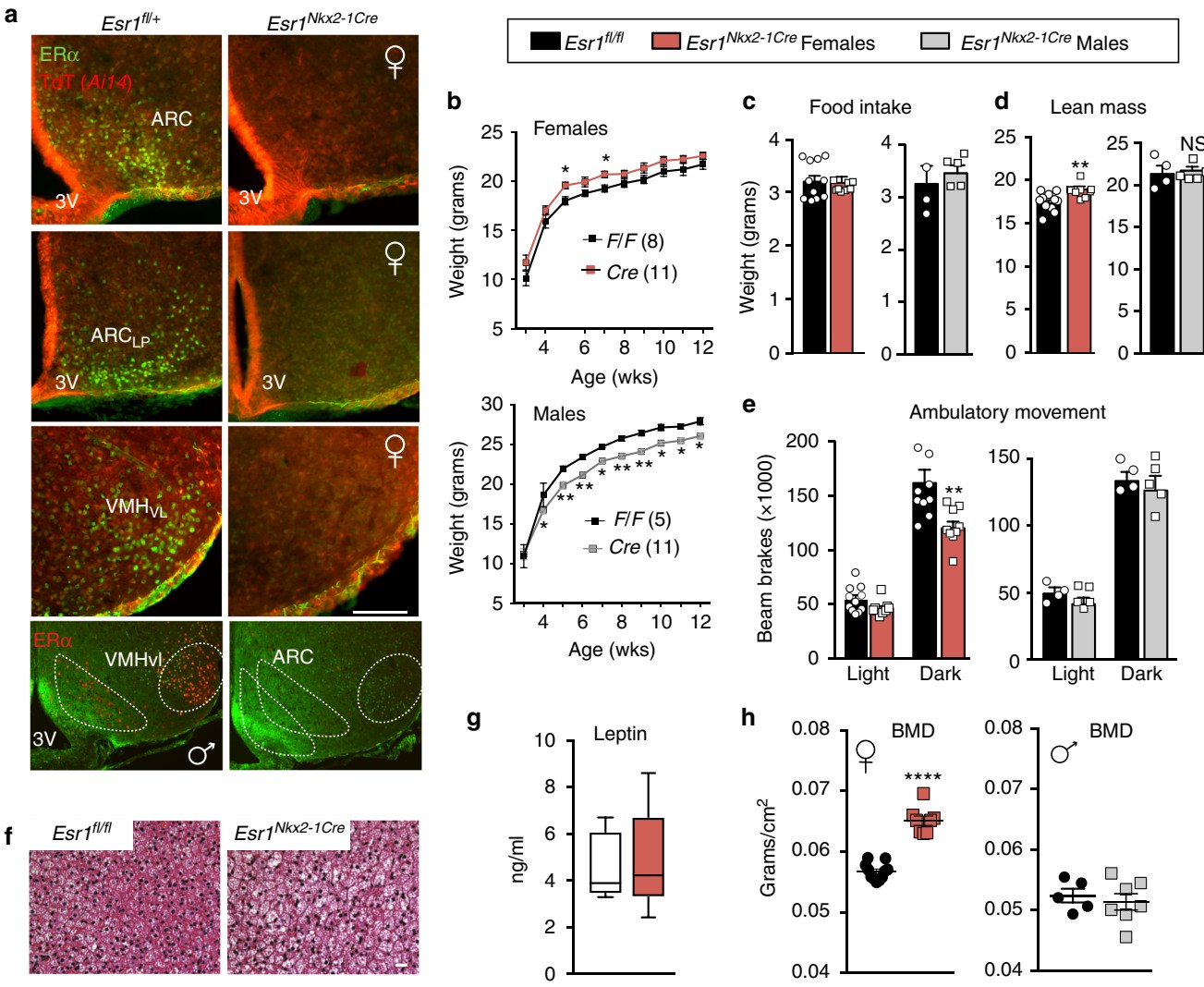

**Fig. 1** Ablating ERα in MBH impairs energy expenditure and increases bone density. **a** Immunohistochemistry of ERα (green) or native TdTomato (TdT (Ai14); red) on coronal brain sections (scale bar = 100 μm) of $Esr1^{fl/+}$; $Ai14^{fl/+}$; $Nkx2$-$1Cre$ (control) or $Esr1^{fl/fl}$; $Ai14^{fl/fl}$; $Nkx2$-$1Cre$ (mutant) from females or males (bottom panels). Lp lateroposterior. Fl/+ = heterozygous for Esr1 floxed allele. **b** Body weight curves of control (black line) or mutant females (red line) ($F_{1,250}$ = 57.01, $p < 0.0001$) and males (grey line) ($F_{3,316}$ = 25.39, $p < 0.0001$) fed on standard chow from 3 weeks of age. **c** Daily food intake per animal over 24 h determined in CLAMS (controls (black bars) mutant females (red bars) and mutant males (grey bars)). **d** Lean mass and **e** averages of ambulatory movement per animal over 12 h determined by metabolic chamber analyses for $Esr1^{fl/fl}$ and $Esr1^{Nkx2-1Cre}$ female and male cohorts, female ambulatory movement ($F$ 1,36 = 10.14, $p = 0.003$). **f** Animals are 8–9-week old. **f** Representative images of hematoxylin and eosin (H&E) staining of BAT $Esr1^{fl/fl}$ and $Esr1^{Nkx2-1Cre}$ females housed at 22 °C (8–16 weeks). Scale bar = 100 μm. **g** Serum leptin levels in 9-week-old control (open bars bar, $n = 5$) and mutant females (red bar, $n = 5$) (center line = median; bounds extend minimum to maximum). **h** BMD measured by DEXA in $Esr1^{fl/fl}$ (black circles) and $Esr1^{Nkx2-1Cre}$ females (red squares, 16–23 weeks) and males (grey squares, 11–18 weeks). Unless otherwise indicated, number per group for female controls ($n = 11$) and mutants ($n = 9$) and for male controls ($n = 4$) and mutants ($n = 5$). Error bars are SEM. Two-way ANOVA (**b**, **e**). Unpaired Student's $t$ tests (**c**, **d**, **g**, **h**). For all figures, $p$ values = $^{*}p < 0.05$; $^{**}p < 0.01$; $^{***}p < 0.001$; $^{****}p < 0.0001$. NS = $p > 0.05$

in juvenile $Esr1^{Nkx2-1Cre}$ males by castration failed to elevate their bone mass, implying that male gonadal hormones are unable to account for the lack of high bone mass in $Esr1^{Nkx2-1Cre}$ males (Supplementary Figure 3D). These data imply that while the high BMD in $Esr1^{Nkx2-1Cre}$ females is partially maintained by ovarian steroids, elevated levels of circulating E2 or pituitary hormones are not the primary drivers of this sex-dependent bone phenotype.

Mechanical bone strength tests established that femora and L5 vertebrae in older $Esr1^{Nkx2-1Cre}$ females were substantially stronger than controls (Fig. 2e). The dense skeletal phenotype in $Esr1^{Nkx2-1Cre}$ females observed in femoral and vertebral trabecular bone emerged early and continued to persist in older

females (54–74 weeks), exceeding values found for OVX mutant females (Fig. 2c, f, g). Thus, trabecular bone, which becomes porous and more fragile in osteoporosis, is remarkably dense and durable in older $Esr1^{Nkx2-1Cre}$ females. Upregulation of bone metabolism in $Esr1^{Nkx2-1Cre}$ females was not associated with ectopic Cre expression in femoral bone (Supplementary Figure 3F). Representative H&E stained femoral bone sections from juvenile female mice illustrate the striking increase in bone density accompanied by a marked decrease in bone marrow space (Fig. 2h). Despite a narrowing of the bone marrow cavity, no differences in spleen weights were observed in mutant females compared to controls at all ages examined (Supplementary Figure 3E).

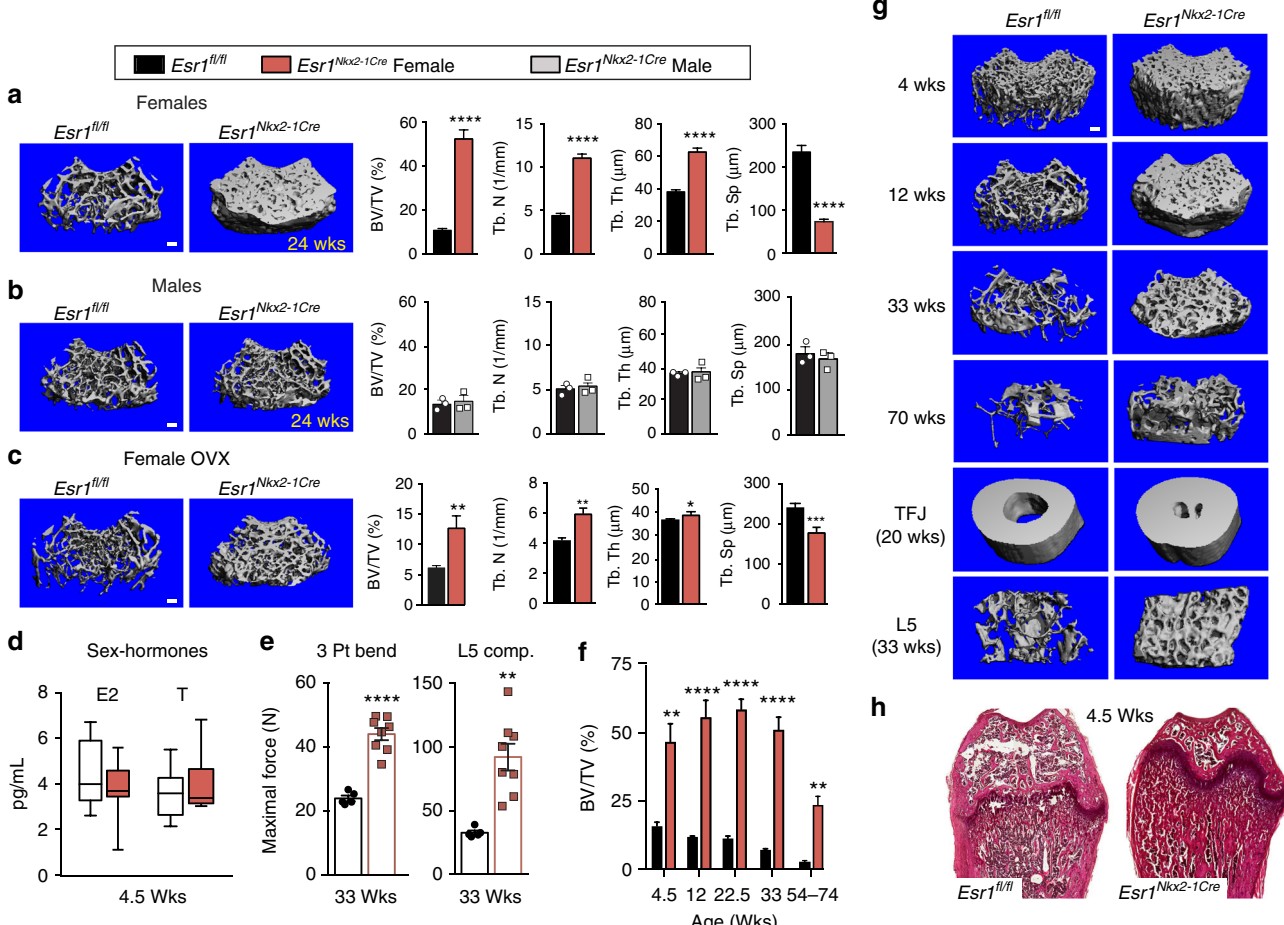

**Fig. 2** Sex-dependent increase in bone mass and strength in *Esr1^Nkx2-1Cre* females. Representative μCT 3D reconstruction images of distal femurs in ~24-week-old *Esr1^fl/fl* and *Esr1^Nkx2-1Cre* **a** females, **b** males, and **c** OVX females (15–21 weeks). Right panels show quantitative morphometric properties of distal femurs showing fractional bone volume (BV/TV (%)); trabecular number (Tb. N) trabecular thickness (Tb. Th), and separation (Tb. Sp). For females ($n =$ 11) controls and ($n = 8$) mutants and for males ($n = 3, 3$). For OVX ($n = 11$) controls and ($n = 8$) mutants. *Esr1^fl/fl* (black boxes), *Esr1^Nkx2-1Cre* females (red bars) and males (grey bars). **d** LC–MS/MS of plasma E2 and T for *Esr1^fl/fl* ($n = 9$, open box) and *Esr1^Nkx2-1Cre* ($n = 11$, red box) juvenile females at 4.5 weeks of age (center line = median; bounds extend minimum to maximum). **e** Scatter plot of mechanical testing of distal femurs and L5 vertebral bodies (33 weeks) *Esr1^fl/fl* (black squares), *Esr1^Nkx2-1Cre* (red squares). **f** BV/TV (%) of the distal femur generated by either μCT or 2D histomorphometric analysis over time from 4.5 weeks of age to 54–74 weeks of age, for genotype ($F_{1,50} = 172.1$, $p < 0.0001$), animal number in each *Esr1^fl/fl* and *Esr1^Nkx2-1Cre* group for 4.5 weeks ($n = 3, 4$), 12 weeks ($n = 5, 6$), 22.5 weeks (4, 5), 33 weeks (7, 9), and 54+ weeks (9, 7). **g** Representative μCT images of age-dependent changes in femoral bone mass, as well as image of cortical bone at the tibial fibular joint (TFJ) in females (20 weeks), and L5 vertebral bodies (33 weeks) in *Esr1^fl/fl* and *Esr1^Nkx2-1Cre* females. **h** Representative H&E staining of female femurs from *Esr1^fl/fl* and *Esr1^Nkx2-1Cre* juvenile females at 4.5 weeks. Error bars are ±SEM (standard error of the mean). Two-way ANOVA (**e**), Unpaired Student's *t* test (**a–d**, **f**). *$p < 0.05$; **$p < 0.01$; ***$p < 0.001$; ****$p < 0.0001$. Scale bars = 100 μm

**Elevated bone formation rate in *Esr1^Nkx2-1Cre* female femurs.** Young *Esr1^Nkx2-1Cre* females showed a significant increase in bone formation rate (BFR) and mineralized surface (Fig. 3a, b), demonstrating robust osteoblast function. Both the mineral apposition rate (MAR) and normalized osteoclast number were unaffected in mutant bone, implying that significant decreases in osteoclast number and function are unable to account for the high bone mass phenotype (Fig. 3b and Supplementary Figure 3G). A similar trend was observed in older *Esr1^Nkx2-1Cre* females after maximal bone density is achieved (Supplementary Figure 3G). Based on transcriptional profiling, gene changes in mutant bone marrow included upregulation of BMP signaling and osteoblast differentiation/ossification (Supplementary Dataset 1) with a concomitant elevation of *Sp7* (*Osterix*), *Wnt10b*, *Bglap* (*Osteocalcin*), *Sost*, and osteoclast markers in mutant bone (Fig. 3c, d). While *Runx2* was unchanged in mutant bone chips harvested from long bones minus endplates, this osteoblast

precursor marker was modestly increased in female bone marrow when examined at 4.5 weeks of age (Fig. 3d). Markers for chondrocyte differentiation (*Sox9*[27]) and those that would indicate a change in sympathetic tone[28,29], were unchanged (Supplementary Table 1 and Dataset 1), whereas markers of interferon (IFN) signaling (*Oas2*, *Oas3*, *Itga11*, *Gbp6*, and *Gbp4*) and cartilage (*Cola2*) were elevated (Fig. 3c, Supplementary Figure 3H and Dataset 1), consistent with the estrogen-independent increases in bone mass following interferon-gamma treatment[30]. Collectively, these data suggest that ablating ERα in the MBH leads to an expansion of bone marrow stromal cells or adult skeletal stem cells[31], fated for osteoblast differentiation/proliferation that give rise to mature bone and cartilage.

**Elevated bone mass after stereotaxic deletion of ERα in ARC.** To unequivocally establish that the high bone mass phenotype in

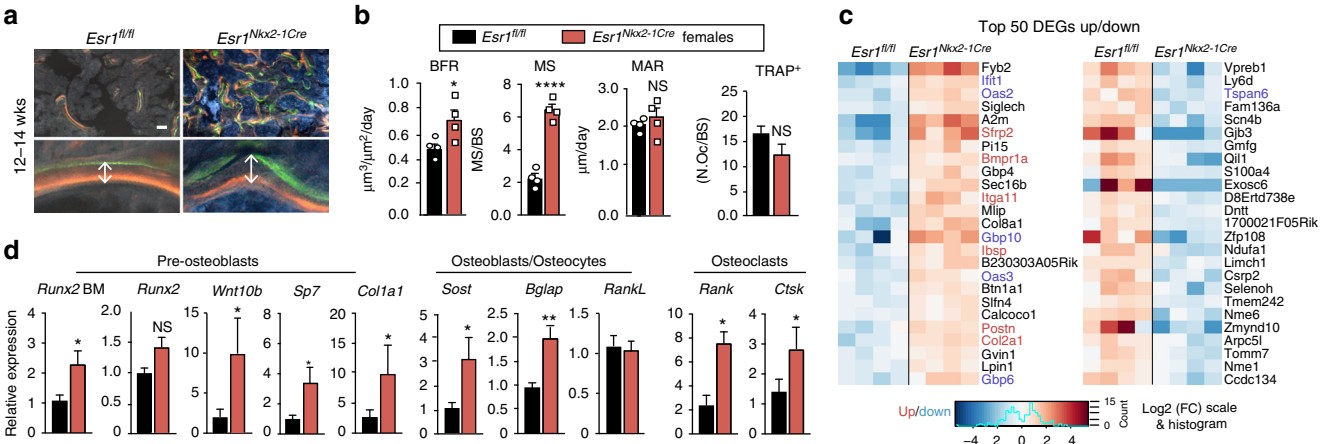

**Fig. 3** Increased bone formation in *Esr1^Nkx2-1Cre* females. **a** Representative images of labeled mineralized surface of distal femur with calcein (green) and demeclocycline (orange) over a period of 1 week in a 12 week old female. Scale bars = 50 μm. **b** Dynamic histomorphometric results for *Esr1^fl/fl* (*n* = 4, black bars) and *Esr1^Nkx2-1Cre* (*n* = 4, red bars) 12–14 week females showing bone formation rate (BFR), mineralized surface (MS), and mineralized apposition rate (MAR). Number of active osteoclasts normalized to bone surface quantified by TRAP-positive staining determined in distal femurs from 5- to 7-week-old *Esr1^fl/fl* (*n* = 5) and *Esr1^Nkx2-1Cre* (*n* = 6) females. **c** Heat map of top 50 differentially expressed genes (DEGs) up and down in 4.5-week-old *Esr1^fl/fl* and *Esr1^Nkx2-1Cre* bone marrow *Esr1^fl/fl* (*n* = 4) and *Esr1^Nkx2-1Cre* (*n* = 4) females. BMP regulated genes (red) and IFN regulated genes (blue). **d** Quantification of indicated transcripts marking pre-osteoblasts, osteocytes, and osteoclasts in 4.5–7-week female control (*n* = 10) and mutant (*n* = 7) flushed bone marrow (BM), or in female control (*n* = 13) and mutant (*n* = 8) femur bone chips. Error bars are ±SEM. Unpaired Student's *t* test (**b**, **d**). *$p < 0.05$; ****$p < 0.0001$. NS = $p > 0.05$

*Esr1^Nkx2-1Cre* females arises specifically from loss of ERα signaling in the brain, stereotaxic delivery of AAV2-Cre was used to eliminate ERα in either the VMHvl or the ARC (referred to as ERαKO^VMHvl or ERαKO^ARC, respectively, Fig. 4a). Adult *Esr1^fl/fl* females injected with either AAV2-GFP (control) or AAV2-Cre were evaluated for ERα expression (Supplementary Figures 4A, B). Successful hits were defined as partial or full loss of ERα on one or both sides of the VMHvl or ARC. As noted for *Esr1^Nkx2-1Cre* females, eliminating ERα in the ARC but not the VMHvl fully recapitulated the significant increase in BMD without changing food intake or E2, T, Leptin, and uterine weights (Fig. 4b–d and Supplementary Figure 5A), disentangling the high bone mass phenotype in ERαKO^ARC females from changes in these circulating hormones. Strikingly, just 12 weeks postinfection (PI), ERαKO^ARC females showed a similar massive elevation in fractional femoral bone volume, which was accompanied by an increase in trabecular number and thickness and decreased bone marrow space (Fig. 4e–g), as well as a modest increase in osteoprotegerin and elevated SOST (Supplementary Figure 5A). Cortical bone thickness was also enhanced without affecting the cortical perimeter (Fig. 4f and Supplementary Figure 5B).

The impact of viral-mediated deletion of ERα in the ARC was assessed over time in older OVX females that partially model postmenopausal bone loss. As expected, in vivo imaging showed that volumetric bone in OVX females dropped rapidly after ovariectomy, dipping by half from ~11 to 5.6 ± 1.3 SEM %BV/TV for all cohorts. Bone mass declined further to ~3.1 ± 0.9 SEM %BV/TV 5-week PI—the time period required to achieve complete ERα deletion in the ARC[32]. While bone density continued to deteriorate in control groups where ERα remained intact (GFP or Miss), complete or partial loss of ERα in the ARC (Hits) resulted in a remarkable ~50% increase in bone volume 12-week PI despite the mature age (38 weeks) of these older females (Fig. 4g and Supplementary Figure 4C). In sum, these data using both intact and older estrogen-depleted females suggest that the increased bone formation observed in *Esr1^Nkx2-1Cre* females is central in origin, thus supporting the existence of a robust estrogen-sensitive neuroskeletal circuit.

**Altered *Kiss1* neuron transcripts in *Esr1^Nkx2-1Cre* females.** To assess molecular changes in the ARC that are associated with upregulation of bone metabolism in *Esr1^Nkx2-1Cre* females, transcriptional profiling was performed. Using microdissected female ARC tissue from controls and mutants, we defined ~180 DEG significantly changed in *Esr1^Nkx2-1Cre* mutants (Fig. 5a, b). Of those transcripts, 83% overlapped with genes that are known to be regulated by estrogen (Fig. 5c) as illustrated by loss of *Greb1*, a highly responsive ERα gene target[33]. Strikingly, however, more than 100 differentially regulated transcripts were distinct from the well-characterized markers of either POMC or AgRP neurons (Fig. 5c and Supplementary Dataset 2). Among significantly downregulated genes, four transcripts are associated with dopaminergic neurons: the dopamine transporter *Slc6a3* (DAT), the synaptic vesicle glycoprotein *Sv2c*, the transcription factor *Nr4a2*, and the prolactin receptor *Prlr* (Fig. 5a). After loss of ERα, *Slc6a3* is downregulated in the ARC consistent with *Slc6a3* upregulation by E2 in cultured cells[34] (Fig. 5d, e). Accordingly, we find that the majority of DAT-positive neurons in the dorsal medial ARC coexpress ERα, by means of an *Slc6a3^Cre;Tdtomato* reporter line (Fig. 5e). Another triad of DEGs is *Kiss1*, *Pdyn*, and *Tac2* (Fig. 5b) that together with the glutamate transporter *Slc17a6* define KNDy (Kisspeptin, Neurokinin B, Dynorphin) ARC neurons[35,36]. As expected and based on the dynamic transcriptional repression of KNDy markers by estrogen[20], both *Kiss1* and *Pdyn* are elevated in *Esr1^Nkx2-1Cre* mutants (Fig. 5d).

**Deleting ERα signaling in *Kiss1* cells elevates bone mass.** Given that the gene signature of KNDy neurons is altered in *Esr1^Nkx2-1Cre* females, we deleted ERα in *Kiss1* cells (*Esr1^Kiss1-Cre*) using a *Kiss1-Cre-GFP* knockin allele to ask if we might identify which estrogen-responsive ARC neurons drive the robust female bone phenotype. As the majority of *Kiss1* neurons share a common lineage with *POMC* neurons in development[37], we also deleted ERα in *POMC* neurons (*Esr1^POMC-Cre*). While ERα was partially ablated in the *Esr1^POMC-Cre* ARC, we failed to detect a trabecular or cortical bone phenotype previously reported for *Esr1^POMC-Cre* females[7] (Fig. 6a and Supplementary Figures 6A, B); our negative results could stem

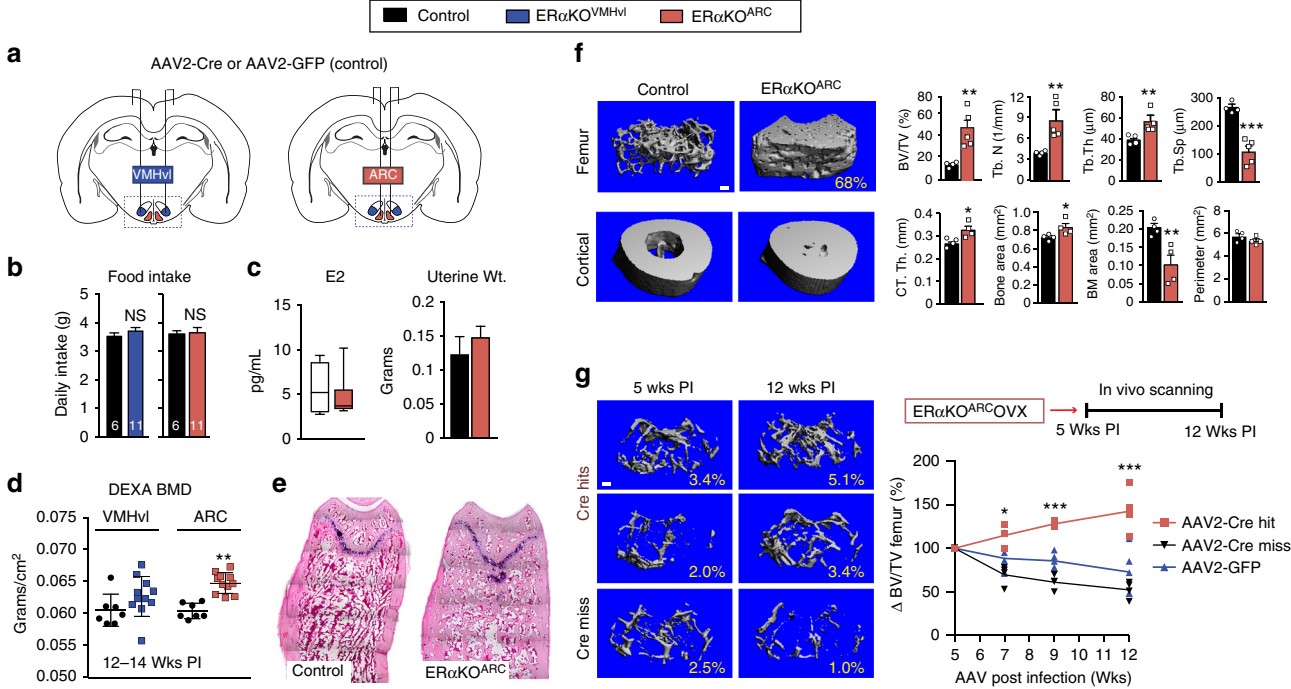

**Fig. 4** Bone volume in intact and OVX female mice after acute loss of ERα in ARC. **a** Schematic of stereotaxic delivery of AAV2-GFP or AAV2-Cre-GFP to 16-week-old *Esr1fl/fl* females to either the VMHvl or ARC regions to delete ERα 5-week postinfection (PI). **b** Food intake for AAV2-GFP (black bars), AAV2-Cre-VMHvl (blue bars) and AAV2-Cre-ARC (red bars). **c** LC–MS/MS plasma E2 and uterine weight for control ($n = 6$, open box) and for ERαKOARC ($n = 9$–11, red box) (center line = median; bounds are minimum to maximum). **d** Scatter plot of BMD for controls, ERαKOVMHvl, and ERαKOARC females. **e** Representative images of distal femur (H&E) in control and ERαKOARC females. **f** μCT images with morphometric properties of distal femur for controls ($n = 4$) and ERαKOARC ($n = 5$) and tibio-fibular joint in control ($n = 4$) and ERαKOARC ($n = 4$) females. Scale bar = 100 μm. **g** Representative μCT images of distal femur in OVX females 5- and 12-week postinfection, showing AAV2-Cre hit to ARC (Cre-Hits, Red line), AAV2-Cre miss to ARC (Cre-Miss, black line and AAV2-GFP to ARC (blue line)). Schematic of time line for in vivo bone imaging from 5- to 12-week PI, with graph showing percent change in volumetric bone normalized to 5 week PI. Cre-Hits ($n = 5$), Cre-Miss ($n = 4$), and GFP ($n = 5$), AAV2-Cre Hit versus GFP or Miss ($F_{2,40} = 56.9$, $p < 0.0001$). Error bars are ±SEM. Student's unpaired *t* test (**d**, **f**). Two-way ANOVA (**g**). $^{*}p < 0.05$; $^{**}p < 0.01$; $^{***}p < 0.001$

from strain, dosage or transmission (paternal versus maternal) differences. In stark contrast, after confirming loss of ERα in all *Kiss1* ARC neurons in *Esr1Kiss1-Cre* females (Fig. 6b and Supplementary Figure 6C), both juvenile and older mutant females displayed a striking increase in bone mass that was easily visualized by the naked eye (Fig. 6c), with values reaching 88% BV/TV for the distal femur; mutant L5 vertebrae and cortical bone mass were similarly affected (Fig. 6d and Supplementary Figure 6D). The striking elevation in bone density in *Esr1Kiss1-Cre* females exceeded the observed bone mass in *Esr1Nkx2-1Cre* females at all ages, perhaps reflecting the narrow versus broad expression of these two different Cre drivers. Similar to *Esr1Nkx2-1Cre* mice the bone phenotype is limited to females and appears to be independent of high E2 levels (Fig. 6d and Supplementary Figure 6E), consistent with the findings that deleting ERα with other *Kiss1-Cre* knockin alleles accelerates pubertal onset in female mice without altering negative feedback[19,38]. Although E2 levels were unchanged in juvenile *Esr1Kiss1-Cre* at 4.5–19 weeks, it is possible that the higher average volumetric bone mass for the distal femur observed in *Esr1Kiss1-Cre* compared to *Esr1Nkx2-1Cre* females ($79 \pm 4.8$ versus $52 \pm 4.5$ %BV/TV) results from the premature postnatal LH surge, as noted by others[19,39]. As might be predicted with extremely dense bone and probable bone marrow failure, spleen weights and markers of extramedullary hematopoiesis (indicated by megakaryocytes) increase significantly in females (Fig. 6e and Supplementary Figure 6F). Taken together, the contrasting bone phenotypes observed in *Esr1Kiss1-Cre* and *Esr1POMC-Cre* infer that this female-specific brain-to-bone pathway is mediated by a subset of *Kiss1* neurons that arise independently of the POMC lineage[37]. Collectively, our data also suggest that disrupting the

transcriptional output and activity of KNDy neurons breaks a brain–bone homeostatic axis that would normally restrain anabolic bone metabolism.

## Discussion

Our investigation to understand the complex role of estrogen signaling in the MBH establishes that ERα-expressing *Kiss1* ARC neurons are central to restraining a powerful brain–bone axis in female mice. This assertion stems from the sex-dependent, high bone mass phenotype that emerged from three independent, intersectional strategies that target central ERα signaling. When compared with other mouse models that alter bone remodeling, several prominent features emerge from our results. In particular, the only model that, to our knowledge, rivals the magnitude of volumetric bone density increase observed in *Esr1Kiss1-Cre* and *Esr1Nkx2-1Cre* females is the sclerostin null (*Sost−/−*) mouse[40,41]. However, the *Sost−/−* bone phenotype is observed in both sexes and the connectivity density is substantially lower[40]. Moreover, we find that selectively removing ERα in the ARC of older, estrogen depleted females results in an impressive ~50% increase in bone density, indicating a potential therapeutic value in manipulating this female neuroskeletal circuit. Disrupting this neuroskeletal circuit enhances genetic pathways associated with osteogenesis and results in fully functional mature bones with exceptional strength. When considered alongside the well-established role of peripheral estrogen in the prevention of bone loss[42], our findings illustrate that the same hypothalamic neurons used to restrain the onset of puberty also inhibit anabolic bone metabolism in females. We speculate that once this female

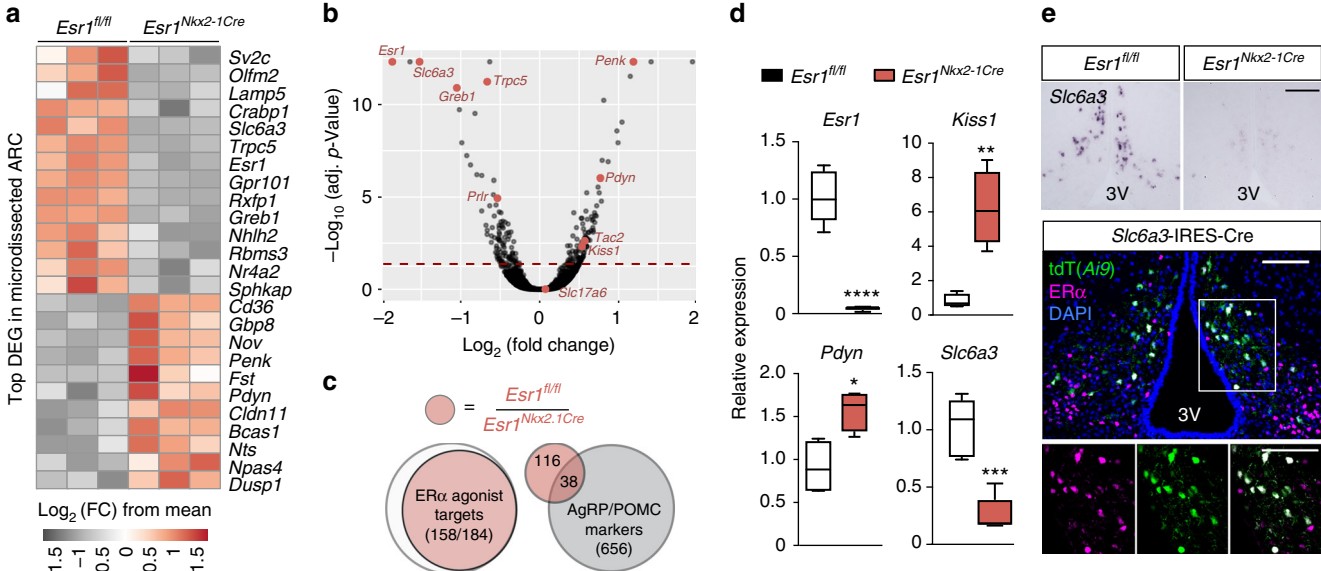

**Fig. 5** Bone formation correlates with changes in KNDy and DAT ARC neurons. **a** Heat map of top 25 most significant DEGs in ARC of *Esr1^fl/fl* (*n* = 3) and *Esr1^Nkx2.1Cre* females (*n* = 3) (15 weeks). **b** Volcano plot of data set with highlighted genes (red). Dashed red line represents significance cutoff with adjusted *p* value < 0.05. **c** Overlap of DEGs with adjusted *p* value < 0.05 and |log2 fold change| > 0.6 between ARC from *Esr1^fl/fl* and *Esr1^Nkx2.1Cre* mice (red) with identified ERα agonist-responsive transcripts (white)[61,62] and with identified markers of AgRP and POMC neurons (grey)[63]. **d** Expression of *Esr1*, *Kiss1*, *Pdyn*, and *Slc6a3* measured by qPCR (*n* = 4–6 per genotype with controls as open boxes and *Esr1^Nkx2.1Cre* females as red boxes; (center line = median; bounds are minimum to maximum). **e** Representative ISH of *Slc6a3* in ARC and confocal image of ARC co-labeled with *Slc6a3* reporter (*Ai9*, tdT), ERα and DAPI. Image scale bars for top and bottom panels = 100 μm. Student's unpaired *t* test (**d**). Error bars are ±SEM. *p < 0.05; **p < 0.01; ***p < 0.001; ****p < 0.0001

ERα-dependent brain-to-bone pathway is disturbed, energetic resources are funneled into bone and diverted away from reproduction and energy expenditure (Fig. 6f).

*Kiss1* hypothalamic neurons can be categorized as either KNDy in the ARC or Kiss1 in the rostral ARC (AVPV). Multiple labs report that Kiss1 and KNDy subtypes are distinguished by their neuronal excitability[21,22], projections[43], and regulatory function of ERα, including a role of nonclassical, non-ERE action[44]. At this juncture, we know that deleting ERα with a *Kiss1-Cre* knock-in allele, which will target the ARC and AVPV, as well as other tissues, triggers an incredibly robust female high bone mass phenotype. When coupled with a similar phenotype observed in ERαKO^ARC, and the residual ERα expression in the AVPV of *Esr1^Nkx2.1Cre* females, we reason that it is not Kiss1 AVPV neurons, but KNDy ARC neurons that regulate this sex-dependent brain-to-bone connection. Whether there are functionally distinct KNDy ARC neuronal subpopulations critical for this brain-to-bone pathway remains to be determined. It is possible that the source of this brain-to-bone pathway might stem indirectly from KNDy neurons regulating POMC tone, that then influence bone remodeling[45]. Given that prodynorphin, a marker of KNDy ARC neurons is suppressed by estrogen, but not by tamoxifen[46], one might also speculate that some of the bone-sparing effects of this selective ERα modulator[47] stem from its antagonist activity in the ARC.

The precise neuronal or humoral signals that promote the high mass bone phenotype in *Esr1^Nkx2-1Cre*, *Esr1^Kiss1-Cre*, and ERαKO^ARC females remain to be determined. However, we note that this phenotype is independent of changes to leptin or E2 and is not directly influenced by ERα neurons in the VMHvl. In this respect, our results differ from prior reports linking leptin deficiency to high trabecular bone mass[48] via a circuit involving suppression of serotonergic signaling in the VMH[49] or direct effects of leptin on bone[50]. Moreover, while lower sympathetic output in the MBH can lead to mild increases in cortical and trabecular bone over months[51], the high bone density phenotype

in *Esr1^Nkx2-1Cre* and *Esr1^Kiss1-Cre* females arises as early as 4 weeks, and all measured parameters of sympathetic tone (e.g., circulating ACTH, catecholamines, markers of changed sympathetic tone in either bone or BAT) are not significantly different in mutant females (Supplementary Figures 2F, G and Supplementary Figure 3C and Supplementary Table 1). Importantly, the elevated bone mass in mutant females does not appear to be caused from impaired bone resorption as no change in tartrate-resistant acid phosphatase (TRAP) staining was observed in mutant female femoral bone. That sclerostin, a known repressor of bone metabolism is elevated in *Esr1^Nkx2-1Cre* mutants implies that their massive increase in female volumetric bone mass is independent of sclerostin. Thus, we conclude that the high bone mass in our mouse models results from activation of a potent signaling pathway that promotes bone formation by a humoral mechanism and is initiated in the brain.

Our findings raise an interesting question: why have a female-specific ERα brain–bone pathway that counteracts the positive effects of peripheral estrogen on bone remodeling? One clue is provided by the global role of ERα signaling in the MBH, which is to allocate energetic resources to initiate and preserve reproduction. Based on our loss-of-function studies, we conclude that ERα signaling in MBH neurons regulates energy expenditure in this way, by promoting locomotion, generating heat via adaptive BAT thermogenesis, and preserving energy and calcium stores by preventing excessive bone building, without altering food intake. It is important to note that the high bone phenotype degrades far less in older intact mutant females (52% bone loss) compared to OVX mutant females (75% bone loss), as our data show for 54–74-week-old females. Collectively, these data imply that disrupting the brain-to-bone pathway is unable to counteract fully the significant bone loss due to estrogen depletion. However, understanding how *Esr1^Nkx2-1Cre* mutants females are able to preserve much of their bone during aging is clearly relevant to future translation of these basic findings. As such, our work

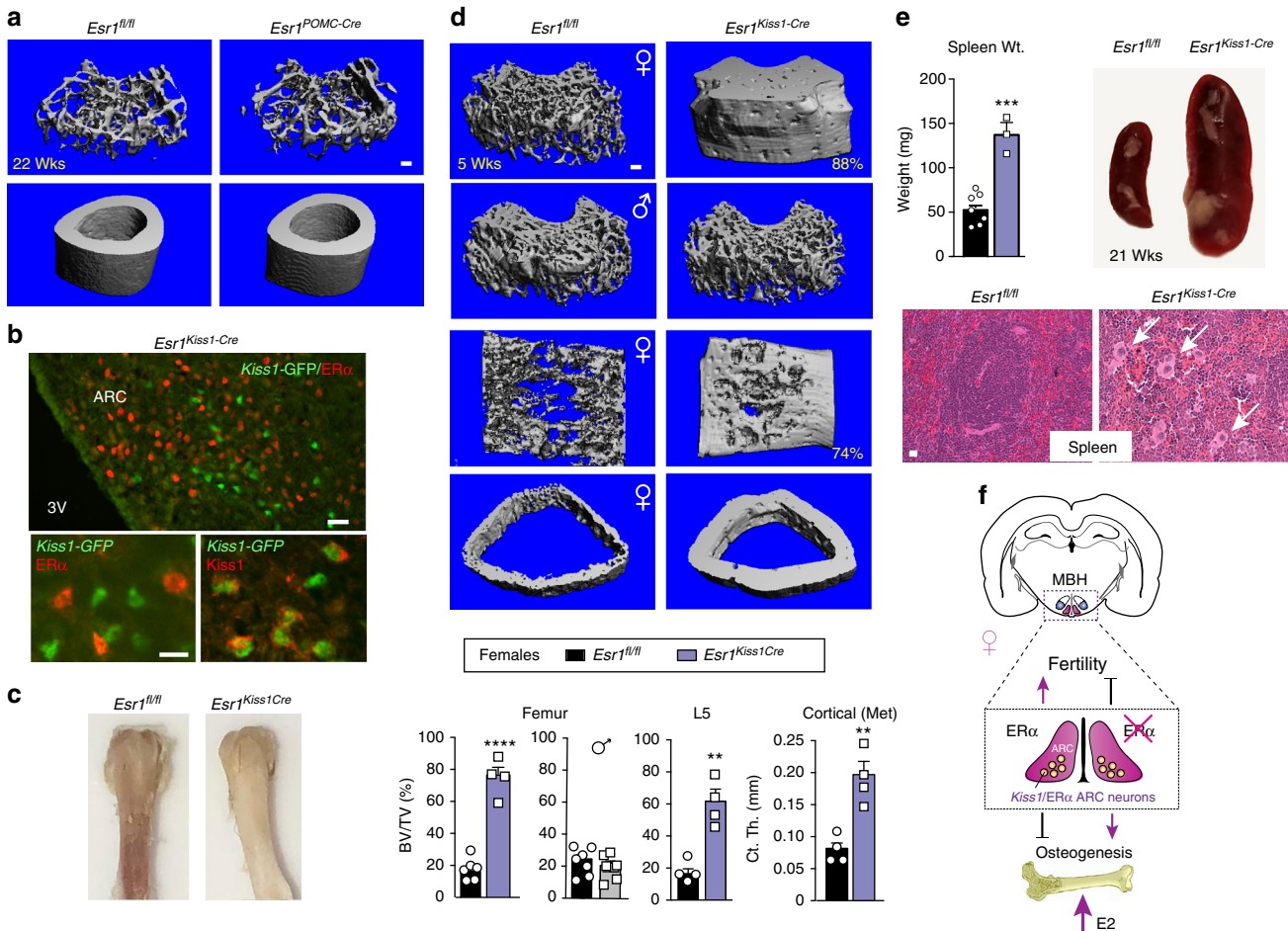

**Fig. 6** Deletion of ERα in *Kiss1* neurons elicits an increase in bone mass. **a** Representative µCT images of femoral and cortical bone for *Esr1*[fl/fl] ($n = 3$) and *Esr1*[PomcCre] ($n = 3$) at 22 weeks of age. **b** Representative images demonstrating loss of ERα (red) in Cre-GFP-expressing *Kiss1* neurons stained for either GFP (green) or KISS1 (red) in the female ARC. Scale bars = 25 µm (top panel) and 10 µm (bottom panels). **c** Photograph of *Esr1*[fl/fl] and *Esr1*[KissCre] female femurs at 6 weeks of age. **d** µCT images of the distal femur, L5 vertebrae and cross section of cortical bone at metaphysis in *Esr1*[fl/fl] (black) control and *Esr1*[KissCre] mutant females (purple) as well as the distal femur from males (grey, 4.5 weeks). %BV/TV for female distal femur controls ($n = 6$) and mutants ($n = 4$), for L5 controls ($n = 4$) and mutants ($n = 4$), and the cortical thickness at metaphysis for controls ($n = 4$) and mutants ($n = 4$). %BV/TV for male distal femur controls ($n = 7$) and mutants ($n = 6$) males. **e** Spleen weights of younger (4–12 weeks) *Esr1*[fl/fl] ($n = 7$) and *Esr1*[KissCre] ($n = 3$) and images of spleen from a control and a mutant female at 21 weeks of age. Representative images of (20×) H&E stained spleens from control and mutant females at 21 weeks. White arrows point to megakaryocytes. **f** Schematic showing the positive and negative roles of central estrogen signaling in ARC[Kiss1] neurons on fertility and bone, respectively. Error bars are ±SEM. Unpaired Student's *t* test (**d**, **e**). $^{**}p < 0.01$; $^{***}p < 0.001$; $^{****}p < 0.0001$. Scale bars = 100 µm

defines central regulation of bone metabolism, alongside reproduction and energy balance, as a fundamental determinant of female physiology. This estrogen-sensitive neuroskeletal axis is likely to be relevant during the prepubertal growth spurt in humans and in late stages of pregnancy, when gonadal steroids are low or high, respectively.

In the course of our study, we also found that *Slc6a3* encoding DAT is highly responsive to estrogen and marks a subset of dorsal medial ERα ARC neurons. Although *Kiss1* neurons appear to be sufficient in mediating the central effects of ERα on bone, defining the contribution of DAT ARC neurons to this circuit awaits development of better genetic tools. Unfortunately, consistent with earlier studies showing limited efficacy of *Slc6a3-Cre* in the hypothalamus[52,53], ERα remained intact in DAT + ARC neurons in *Esr1*[Slc6a3Cre] mice. Because dopaminergic ARC neurons are modulated by Kiss1[54], it will be of interest to determine how these two estrogen-responsive ARC modules communicate to coordinate female bone and energy metabolism before, during, and after pregnancy.

In summary, our work reveals an unprecedented sex-dependent bone phenotype and provides unequivocal proof of brain-to-bone signaling[55]. Furthermore, our findings demonstrate the importance of central estrogen signaling (which exists in a coregulatory system with peripheral estrogen) in the maintenance of bone homeostasis in females. Breaking this neuroskeletal homeostatic circuit in young and old females promotes anabolic bone metabolism and provides a model for further mechanistic investigations that might eventually provide opportunities to counteract age-related osteoporosis in both women and men.

## Methods

**Mice.** The origin of the *Esr1*[fl/fl] allele on a 129P2 background and used to generate *Esr1*[Nkx2-1Cre] mice are described in ref. [11] and were maintained on CD-1;129P2 mixed background. *Esr1*[POMC-Cre] and *Esr1*[Kiss1-Cre] mice were generated by crossing male mice harboring a single copy of the *Pomc-Cre* transgene (official allele: *Tg (Pomc1-cre)16Lowl/J*) or the *Kiss1-Cre-GFP* knockin allele to *Esr1*[fl/fl] females (official allele: *Esr1*[tm1Sakh]). *Pomc-Cre* transgenic mice were obtained from C. Vaisse (UCSF). *Esr1*[POMC-Cre] mice were maintained on a mixed FVB/N, CD-1, 129P2, and C57BL/6 genetic background. The knockin *Kiss-Cre:GFP* (version 2) on

a C57BL/6 background was a gift from R. Steiner and R. Palmiter (UW). $Esr1^{Kiss1-Cre}$ mice were maintained on a mixed C57BL/6; CD-1;129P2 genetic background[56]. $Esr1^{fl/+};Ai14^{fl/+};Nkx2-1Cre$ mice were generated by crossing $Esr1^{fl/+};Ai14^{fl/fl}$ reporter females with $Esr1l^{fl/fl}; Nkx2-1Cre$ males. The $Slc6a3^{Cre};Ai9^{fl/+}$ reporter line was a gift from by R. Edwards (UCSF). Primer sequences used for genotyping can be found in (Supplementary Table 2). For assessment of bone mass in estrogen depleted OVX females, OVX females, ovariectomy was performed between 8 and 25 weeks. For assessment of bone mass in androgen depleted male mice, mice were castrated at 3 weeks of age and bone mass was assessed by MicroCT 4 weeks later. Mice were maintained on a 12 h light/dark cycle with ad libitum access to food and to standard chow diet (5058; LabDiet, 4% fat). All animal procedures were performed in accordance with UCSF institutional guidelines under the Ingraham lab IACUC protocol of record.

**Metabolic analysis**. Comprehensive Laboratory Animal Monitoring Systems (CLAMS) measured $O_2$ consumption, $CO_2$ exhalation, total movement (total beam breaks; X- and Y-axes), ambulatory movement (adjacent beam breaks; X- and Y-axes), and rearing (total beam breaks; Z-axis) at 14 min intervals. Food intake for experiments done with $Esr1^{Nkx2-1Cre}$ mice and female ERαKO$^{VMHvl}$ or ERαKO$^{ARC}$ was determined via CLAMS. Mice were housed in CLAMS for 96 h; the first 24–48 h period of acclimation was not included in analyses. Each experimental cohort VO2 measurements were standardized to lean mass. All CLAMS analyses in females were performed in intact animals. EchoMRI were used to measure body composition. DEXA was used to measure BMD and bone mineral content.

**RNA isolation and qPCR**. Tissue was collected, cleaned of excess tissue and homogenized in TRIzol (Invitrogen). RNA was isolated using chloroform extraction. cDNA was synthesized using the Applied Biosystems High-Capacity cDNA Reverse Transcription Kit. Whole femur samples were cleaned of excess tissue, endplates removed, minced and placed in TRIzol. Bone marrow was flushed from femoral samples using 1% HBSS. RNA was isolated using chloroform extraction for all tissues. cDNA was synthesized with random hexamer primers with the Affymetrix reverse transcriptase kit (Affymetrix). Quantitative polymerase chain reaction (qPCR) expression analysis in femoral samples was performed using SYBR Green. Values were normalized to either *36b4*, *mCyclo*, or *Gapdh*. Sequences for primer pairs can be found in (Supplementary Table 3).

**Rotarod**. Age and weight-matched female $Esr1^{fl/fl}$ and $Esr1^{Nkx2-1Cre}$ mice were tested on a rotating rod (UGO Basile) to assess motor coordination. Prior to data collection mice were trained on how to hold onto the rotating log. At the time of measurement, mice were placed on the rotarod and the speed (rotations/min) gradually increased over a 5-min period. The test ended when the mice fell off the rotarod. The time in seconds at the moment of release was recorded. Mice were subjected to 6 trials over 2 days (3 trials/day).

**Grip strength**. We measured forelimb grip strength using the Grip Strength Meter (Chatillion, Columbus Instruments, OH, USA). Briefly, prior to data collection, mice were trained on holding the instrument grids for grip strength measurement. Animals were held by the tail, directed and allowed to grasp a steel grip gauge with forelimbs only. After gripping the steel grip gauge, mice were gently and steadily pulled away until the grip was released. The force measured at time of grip release was recorded as grip strength. Measurements obtained were calculated as the average of three measurements.

**Immunohistochemistry**. Immunohistochemistry was performed on cryosections (20 µm) collected from brains fixed in 4% paraformaldehyde using standard procedures. ERα staining used rabbit polyclonal anti-ERα (EMD Millipore, Billerica, MA cat # C1355) or mouse monoclonal ERα antibody (Abcam, Cambridge cat # 93021) at a dilution of 1:1000 or 1:100, respectively. GFP staining used anti-GFP antibody (Novus Biologicals, Littleton, CO cat # NB100-1614) at 1:2500. Kisspeptin staining used rabbit polyclonal anti-KISS1 at a dilution of 1:200 (Abcam, Cambridge cat # ab19028). Confocal images were taken using a Nikon Ti inverted fluorescence microscope with CSU-22 spinning disk confocal.

**In situ hybridization**. *Slc6a3* cDNA for in vitro transcription of digoxigenin (DIG)-labeled riboprobe was generated by PCR amplification (forward primer: 5-TTCCGAGAGAAACTGGCCTA-3 and reverse primer: 5-TGTGAAGAGCAG GTGTCCAG-3) from a brain mRNA library. In situ hybridization (ISH) was performed on 20 µm sections using standard protocols[57]. The DIG-riboprobe was hybridized overnight at 65 °C. Following washing and blocking, sections were incubated overnight with anti-DIG-AP (1:2000) (Roche) at 4 °C. AP signal was developed using BM Purple (Roche).

**Serum measurements**. For measurements of E2 or T, nonpolar metabolites from plasma were extracted in 1 ml of PBS with inclusion of internal standards C12:0 monoalkylglycerol ether (MAGE) (10 nmol, Santa Cruz Biotechnology) and pentadecanoic acid (10 nmol, Sigma-Aldrich) and 3 ml of 2:1 chloroform:methanol. Aqueous and organic layers were separated by centrifugation at $1000 \times g$ for 5 min

and the organic layer was collected, dried under a stream of N2 and dissolved in 120 ml chloroform. A 10 µL aliquot was injected onto liquid chromatography/mass spectrometry (LC/MS) and metabolites were separated by LC[58]. MS analysis was performed with an electrospray ionization source on an Agilent 6430 QQQ LC–MS/MS (Agilent Technologies) with the fragmentor voltage set to 100 V, the capillary voltage was set to 3.0 kV, the drying gas temperature was 35 °C, the drying gas flow rate was 10° l/min, and the nebulizer pressure was 35 psi. Metabolites were identified by SRM of the transition from precursor to product ions at associated optimized collision energies and retention times[58]. Metabolites were quantified by integrating the area under the curve, then normalized to internal standard values and values determined by comparison to a standard curve of each metabolite of interest run simultaneous to the experimental samples.

Plasma leptin was measured using the mouse Leptin Elisa Kit (Chrystal Chem). Circulating pituitary hormones, bone markers, and plasma catecholamines were measured by the VUMC Hormone Assay and Analytic Services Core. Briefly, levels of serum pituitary and thyroid hormones were measured by radioimmunoassay. Bone markers were measured by the Millipore bone metabolism mulitplex fluorescent Luminex Assay. For catecholamine analysis, plasma was collected in the afternoon and immediately treated with EGTA–glutathione and subsequently measured by high-performance liquid chromatography.

**Histology**. Tissue was collected and cleaned of excess tissue and fixed in 4% paraformaldehyde and embedded in paraffin. For BAT, ovary and spleen 5 µm sections were cut, processed, stained with hematoxylin and eosin (H&E) and bright field images were taken using the Luminera Infinity-3. Femoral samples were cleaned of soft tissue, fixed in 4% PFA and demineralized in 10% EDTA for 10–14 days before being embedded in paraffin wax. Sections measuring 5 µm were then cut using the Leica RM2165 and subsequently stained with H&E or stained with TRAP. Photoshop software was used to remove background in nontissue areas for images taken of ovaries and distal femurs.

**Stereotaxic delivery of AAV2**. Adeno-associated virus sereotype 2 (AAV2) from UNC Vector Core was injected bilaterally into isoflurane-anesthetized 9–16-week-old adult $Esr1^{fl/fl}$ female mice. VMH coordinates: A–P: −1.56 mm from Bregma; lateral: ±0.85 mm (millimeters) from Bregma; D–V: −5.8 mm from the skull; ARC coordinates: A–P: −1.58 mm from Bregma; lateral ±0.25 mm from Bregma; D–V: −5.8 mm from the skull. AAV2-virus was injected bilaterally into adult $Esr1^{fl/fl}$ 19–24 week-old-female mice 5–8-week post-OVX. Buprenorphine (0.1 mg/kg i.p.) was provided as analgesia after surgery and as needed. For animals receiving AAV2-Cre, an n of at least 12 was used to ensure a large enough sample size considering anticipated misses or miss-targeting of AAV2 virus. Misses were characterized as no ablation of ERα in the VMH or ARC in AAV2-Cre groups.

**Micro-computed tomography**. Volumetric bone density and bone volume were measured at the right femur, tibio-fibular joint, or midshaft, and L5 vertebrae using a Scanco Medical µCT 50 specimen scanner calibrated to a hydroxyapatite phantom. Briefly, samples were fixed in 10% phosphate-buffered formalin and scanned in 70% ethanol. Scanning was performed using a voxel size of 10 mm and an X-ray tube potential of 55 kVp and X-ray intensity of 109 µA. Scanned regions included 2 mm region of the femur proximal to epiphyseal plate, 1 mm region of the femoral mid-diaphysis and the whole L5 vertebrae.

Longitudinal live animal µCT imaging was performed using a Scanco Medical vivaCT 40 preclinical scanner. Animals were anesthetized and a 2 mm region of the distal femur was scanned. Similarly, a trabecular bone compartment of 1 mm length proximal to the epiphyseal plate was measured. Cortical parameters were assessed at the diaphysis in an adjacent 0.4 mm region of the femur.

In both specimen and in vivo scanning, volumes of interest were evaluated using Scanco evaluation software. Representative 3D images created using Scanco Medical mCT Ray v4.0 software. Segmented volume of trabecular bone presented from anterior perspective and ventral side of vertebrae. Segmented volume of cortical bone presented as transverse cross-sectional image. For image acquisition of whole femur, bones were dissected, cleaned of any soft tissue, imaged with iPhone 8 and then cropped and edited in Photoshop CC.

**Biomechanical strength tests**. Right femurs underwent three-point bend test using the ElectroForce 3200 mechanical load frame. Lower supports were separated by a span of 8 mm to support two ends of the specimen. The testing head was aligned at the midpoint between the supports. Femurs were preloaded to a force of 1 N then loaded at a rate of 0.2 mm/s. Loading was terminated upon mechanical failure, determined by a drop in force to 0.5 N. Force displacement data collected every 0.01 s. Load-to-failure tests were performed to measure the uniaxial compressive strength of the L5 vertebral bodies. Before testing, the posterior elements and endplates were removed from the vertebrae, resulting in vertebral bodies with plano-parallel ends. Tests included five preconditioning cycles to 0.3% strain followed by a ramp to failure at a rate of 0.5% strain/sec. Vertebral strength was defined as the maximum compressive force sustained during the tests. All tests were performed at room temperature using an electro-mechanical load frame (Electroforce 3200; Bose, Eden Prairie, MN).

**Dynamic histomorphometry.** To determine bone formation and mineralization, females (age 12–14 and 33 weeks) were injected with 20 mg/kg of calcein (Sigma-Aldrich, St. Louis, MO, USA) 7 days before euthanasia along with 15 mg/kg of demeclocycline (Sigma-Aldrich) 2 days before euthanasia. Bones were fixed in 4% formalin. Before histomorphometric analysis, mosaic-tiled images of distal femurs were acquired at ×20 magnification with a Zeiss Axioplan Imager M1 microscope (Carl Zeiss MicroImaging) fitted with a motorized stage. The tiled images were stitched and converted to a single image using the Axiovision software (Carl Zeiss MicroImaGing) prior to blinded analyses being performed using image-analysis software (Bioquant Image Analysis Corp., Nashville, TN, USA). The dynamic indices of bone formation within the same region that were measured on 10-mm sections and percent mineralizing surface (MS/BS), MAR, and surface-based BFR/BS were determined by Bioquant OSTEO software.

**Microdissection and profiling of the ARC and bone marrow.** Flushed bone marrow was obtained as described above from control and mutant female femur at 4.5 weeks of age. Microdissected ARC tissue was obtained from control and mutant female mice (11–20-week old) using the optic chiasm as a reference point, a 2 mm block of tissue containing the hypothalamus was isolated with a matrix slicer. Microdissection techniques were validated by enrichment for *AgRP, Cited* 1 (ARC), and absence of *Tac1, Cbln1* (VMH). For both bone marrow and ARC, total RNA was purified using the PureLink RNA Mini Kit (Invitrogen, Waltham MA). Amplified cDNA was generated from 10 to 50 ng of total RNA using the Ovation RNA-Seq System V2 or Trio (Nugen, San Carlos, CA). For the ARC, cDNA was fragmented to 200 bp using a Covaris M220 sonicator (Covaris, Woburn, MA). For the ARC, barcoded sequencing libraries were prepared from 100 ng of fragmented cDNA using the Ovation Ultralow System V2 (Nugen, San Carlos, CA) and single-end 50 bp reads sequenced from the multiplexed libraries on the HiSeq 4000 (Illumina, San Diego, CA) at the UCSF Center for Advanced Technologies. For ARC and bone marrow samples, sequencing generated reads were mapped to the mouse genome (NCBI37/mm10) using TopHat2. Reads that mapped to exons in annotated genes were counted using HTSeq[59]. Final quantification and statistical testing of differentially expressed genes (adjusted $p$ value < 0.05) was performed using DESeq2[60].

**Reporting summary.** Further information on experimental design is available in the Nature Research Reporting Summary linked to this article.

## Data availability
All data supporting this publication are available from the authors upon reasonable request. A reporting summary for this article is available as a Supplementary Information file. Deep sequencing data that support the findings of this study are archived under GEO accession numbers [GSE122055 and GSE122291]. Mouse lines may be made available upon request.

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

## Acknowledgements

We wish to thank Drs. R. Edwards, S. Khan, E. Hsiao, C. Paillart, Y. Lin, R. Steiner, R. Palmiter, and A. Xu for reagents, discussions as well as H. Escusa, H.C. Cain, and A. Matcham for assistance with data acquisition and also M. Horwitz at the Gladstone Institute Histology Core. This research was supported by grants to H.A.I. (R01 DK099722, UCSF Women's Reproductive Health RAP Award, and NRSA NDSP P30-DK097748), S.M.C. (K01 DK098320) and UCLA Women's Health Center and CTSI (NIH UL1TR001881), C.B.H. (F32 DK107115-01A1 and AHA Postdoctoral Fellowship 16POST29870011), W.C.K. (AHA Postdoctoral Fellowship 16POST27260361), R.A.N. (VA Merit Review Grant 1l01BX003212), A.F. (NIH P30 AR066262), J.R.B. (K08 DK106577) and D.K.N. (NIH/NCI R01CA172667). We acknowledge the UCSF DERC (NIDDK P30 DK063720), the UCSF CCMBM (NIH P30 AR066262) and the Vanderbilt Hormone Assay Core (NIH DK059637 and DK020593).

## Author contributions

C.B.H. and W.C.K. designed experiments, analyzed data, and wrote the paper. B.F. performed mass spectrometry analysis of mouse serum. L.W. performed histomorphometric and data analyses. J.R.B. performed immunohistochemistry and imaging on coronal brain sections. A.L. performed microCT, three-point bend test and tissue embedding of femurs. Z.Z. and M.S.R. performed immunohistochemistry and subsequent imaging on coronal brain sections. A.F. performed L5 vertebral crush test. M.S. performed RNA-seq analysis on bone marrow. D.K.N. provided expertise in mass spectrometry metabolic analysis. R.A.N. helped design experiments and provided expertise in bone biology. S.M.C. designed experiments, provided animal models and analyzed data. H.A.I. designed experiments and wrote the paper.

## Additional information

**Competing interests:** The authors declare no competing interests.

**One Sentence Summary:** Disrupting central estrogen signaling exposes a powerful sex-dependent brain-to-bone pathway in female mice triggering massive increases in bone density.

