## [Peer Review File · Nature Communications]

Reviewers' comments:

Reviewer #1 (Remarks to the Author):

Manuscript #NCOMMS-18-14301

Herber et al., Estrogen Signaling in Arcuate Kiss1 Neurons Suppresses a Sex-Dependent Circuit That Promotes Dense Strong Bones in Female Mice

The authors present data demonstrating selective deletion of ER α in the mediobasal hypothalamus, in the arcuate nucleus, and in arcuate KNDy neurons alters bone density, remodeling, and gene expression in female but not male mice. The primary conclusion is that activation of ER α in the arcuate KNDy neurons leads to the downstream negative control of bone deposition in an apparent effort to shift energetic costs from bone remodeling to reproduction. While others have demonstrated that deletion of ER α in the MBH, in the arcuate, and in POMC neurons alters bone density in females, this is the first study to identify KNDy neurons as primary mediator of the CNS-driven control of bone physiology in females. As such this manuscript is of interest to multiple fields especially those focused on kisspeptin's role in physiology and general bone physiology. This manuscript is also exciting because it generates a little controversy (POMC vs. Kiss) while opening up a new area to investigate – the unknown KNDy-mediated neurocircuit that modulates bone remodeling, the unknown hormone or neurotransmitter that directly controls bone remodeling, etc. The manuscript is easy to read and understand with a clear description of the experimental design and the results. I have a few criticisms and minor comments that should be addressed prior to acceptance for publication.

Criticisms/Questions:

1. The Nkx2-1-Cre mouse model eliminates expression of the targeted gene not only in the hypothalamus/telencephalon but also in the pituitary, thyroid gland, and lungs. In particular the potential loss of ER α -signaling in the pituitary and thyroid are of concerns as both glands are involved in bone remodeling. The authors either need to confirm that there was minimal loss of ER α in both glands in the Esr1/Nkx2-1Cre KO, assess the impacts of the deletion on respective hormone levels, and/or discuss the potential impacts this may have on bone physiology. The authors did assess pituitary hormones levels in the ER α KO-ARC mouse model (Figure S5). Similar measurements should be conducted in the Esr1/Nkx2-1Cre mouse model.
2. The authors should clearly state that the Esr1/Kiss1-Cre mouse model did not fully recapitulate the Esr1/Nkx2-1Cre mouse model which indicates that KNDy neurons are not the only ER α -expressing neurons involved in this brain-bone circuit. The authors suggested as such with their discussion of ER α -expressing DAT neurons.
3. Did the Esr1/Nkx2-1Cre mouse also exhibit "probable bone marrow failure" as evidenced by an increase in spleen weights found in the Esr1/Kiss1-Cre mouse?
4. Page 5, line 10 – change to "circulating catecholamines were not lower in the mutant." If the data is not significant than the levels are not lower.
5. Page 5, line 5 – How old are these OVX females? It would be helpful to the reader to state the ages for every experiment clearly as there are multiple ages used throughout leading to confusion.
6. 4.5-week-old females are peripubertal and should be described as such or at least juvenile throughout the text. I find juvenile used twice.
7. Due to the rather extreme density in bones, did the authors observe in differences in muscle strength or general mobility beyond the decrease in nighttime locomotor behavior in the KO (Fig. 1E)?

8. In Figure 2E, samples sizes are states as 2 for *Esr1/fl* at 12 weeks and for *Esr1/Nkx2-1Cre* at 54 weeks. How can you then have a valid SEM error bar and statistical analysis with an ANOVA on a sample size of 2?

9. A final point, POMC and KNDy neurons talk to each other (Fu & van den Pol, 2010, *J Neurosci*; Nestor et al 2016 *Mol Endocrinol*). One potential pathway is KNDy neurons modulate POMC tone and thus altering the hindbrain circuit upstream of bone remodeling all under the control of ER α activation in KNDy

Reviewer #2 (Remarks to the Author):

In this paper, Herber et al. demonstrate that central ER α signaling in arcuate *Kiss1* neurons inhibits bone mass in female, but not male, mice. Overall, this is an impressive amount of data and the findings generally support the conclusions. There are, however, several issues for the authors to address:

1. It is somewhat unclear if the central *Kiss1* effect is ligand (E2) dependent or independent? Can the authors clarify this issue?
2. The remarkable sex difference is striking. Is this due to sex steroids or other factors? For example, is there an effect of *Kiss1* or *Nkx2 Cre* deletion in male mice gonadectomized well before puberty (eg at 2-3 weeks of age)? Do the authors have an explanation for this marked sex difference?
3. A significant concern is that the mice used in the studies are in a very heterogeneous background, which can have significant effects on bone. Can the authors perform at least one of the key experiments (eg, *AAV2 Cre*) in a pure background (eg B6) to alleviate this concern?
4. The biological relevance of this central estrogen pathway for bone is still unclear. Perhaps the authors could provide a summary diagram showing how they believe this circuit regulates energy and bone metabolism. How important is this pathway when loss of estrogen consistently leads to low, and not high, bone mass? Clearly the peripheral actions of estrogen on bone are dominant. This should be acknowledged and discussed.
5. The authors are perhaps too dismissive of sympathetic outflow. They do find that the KO mice have a 50% reduction in NE (Fig. S2E) and that *bAR2* is 50% lower (Table S1) in the KO mice. Lack of statistical significance here is likely just a matter of sample size. A more complete assessment of additional *bAR* target genes as described in the papers from the Karsenty lab should be performed.
6. *Nkx2-1* is also expressed in the thyroid. Was ER α deleted in the thyroid and were thyroid hormone levels altered in the KO mice?
7. I realize the authors are enthusiastic about their work, but they should refrain from using words like "astonishing" or "incredible", etc to describe their findings. Please let the reader draw their own conclusions.
8. Fig. 2C – it is clear that the KO mice are not protected against ovx induced bone loss. It seems from Fig 2 that on a percentage basis, they lose as much bone or more than WT mice. These data should be shown (ie, percentage/absolute bone loss following ovx vs sham operated mice). Lack of protection from ovx bone loss would also argue that the peripheral effects of estrogen remain dominant, a point the authors tend to gloss over.
9. Fig 4C – similar issue. Sham operated mice need to be included here to show the magnitude of the bone loss in each ovx group relative to sham.

General Comments:

We wish to thank both reviewers for their enthusiastic responses to our work and constructive comments to improve the rigor and impact of our paper. Indeed, motivated by suggestions made by both reviewers, our manuscript has improved significantly. Importantly, we provide new data demonstrating that the high bone mass phenotype in *Esr1^{Nkx2-1Cre}* mutant females is uncoupled from changes in ER α expression in other tissues (pituitary and thyroid) or from changes in circulating pituitary/thyroid hormones or catecholamines. We also show that despite increased bone mass and lean mass, all indices of muscle strength and general mobility are unchanged in *Esr1^{Nkx2-1Cre}* mutant females. Finally, we eliminated the interesting hypothesis posited by Rev2 that changes in circulating androgens in juvenile male mice might mask a high bone mass phenotype, thus accounting for sex-differences in the phenotype. Other concerns have been addressed by new data or changes to the text. Detailed point-by-point responses to each reviewer are provided below.

Review #1

1. The *Nkx2-1-Cre* mouse model eliminates expression of the targeted gene not only in the hypothalamus/telencephalon but also in the pituitary, thyroid gland, and lungs. In particular the potential loss of ER α -signaling in the pituitary and thyroid are of concerns as both glands are involved in bone remodeling. The authors either need to confirm that there was minimal loss of ER α in both glands in the *Esr1/Nkx2-1Cre* KO, assess the impacts of the deletion on respective hormone levels, and/or discuss the potential impacts this may have on bone physiology. The authors did assess pituitary hormones levels in the ER α KO-ARC mouse model (Figure S5). Similar measurements should be conducted in the *Esr1/Nkx2-1Cre* mouse model.

Rev1 correctly points out that the *Nkx2-1Cre* driver is not restricted to the MBH and would potentially alter ER α expression in other tissues, notably in anterior pituitary (thyrotropes), thyroid gland, and lung. To eliminate the possibility that loss of ER α signaling in these non-CNS tissues might contribute to the high bone mass phenotype, RT-qPCR was used to show that ER α transcripts were not significantly affected in pituitary and thyroid glands in mutant females, suggesting that either ER α expression is low or that the Cre-driver fails to delete ER α in these peripheral *Nkx2-1*-expressing tissues (Fig S1B). As also suggested, we have now included a comprehensive pituitary hormone panel for *Esr1^{Nkx2-1Cre}* females as was originally done for acute knockout females (ERKO^{ARC}). No changes in circulating pituitary or thyroid hormones were observed in either young (7-8 wks) or older (33-71 wks) mutant females compared to littermates (Fig S3C). FSH levels are unchanged in mutant

females, suggesting that our bone phenotype is distinct from a recently reported FSH mechanism (blocking antibodies) that modestly increases bone mass (Liu et al, 2017 *Nature*). These data coupled with our other findings support our assertion that the high bone mass phenotype is central in origin and is uncoupled from changes in pituitary, thyroid, leptin or gonadal hormones.

2. The authors should clearly state that the *Esr1/Kiss1-Cre* mouse model did not fully recapitulate the *Esr1/Nkx2-1Cre* mouse model which indicates that KNDy neurons are not the only ER α -expressing neurons involved in this brain-bone circuit. The authors suggested as such with their discussion of ER α -expressing DAT neurons.

Rev1 correctly points out that we did not adequately discuss the observation that the bone phenotype in *Esr1^{Kiss1Cre}* mutant female mice is higher and more pronounced than observed for *Esr1^{Nkx2-1Cre}* females. We hypothesize that the higher bone mass observed in *Esr1^{Kiss1Cre}* mutant females (BV/TV = 80%) compared to *Esr1^{Nkx2-1Cre}* (BV/TV = 60%) females could reflect the highly restricted expression of the *Kiss1Cre*-driver compared to the *Nkx2-1Cre* driver (Xu Q et al., *J Comp Neurol* 2008 and Steiner and Palmiter UW, personal comm.). Thus, it is possible that the broad expression of *Nkx2-1Cre*-driver throughout all ARC neurons (and other MBH regions) might influence estrogen signaling in other neurons that interface with ARC^{Kiss1} neurons to alter their output. After inspecting spleen histology more closely, we find evidence of extramedullary hematopoiesis including megakaryocyte (precursor cells to blood platelets) infiltrating the spleens of *Esr1^{Kiss1Cre}* female mutants by 21 wks of age – these data are included in a revised Fig 6E. No spleen phenotype has been observed for young (8 wks) and older (70 wks) *Esr1^{Nkx2-1Cre}* females (Fig S3E), most likely reflecting the lower bone mass in these mutant females. The result/discussion sections have been modified to reflect these new data and speculation.

3. Did the *Esr1/Nkx2-1Cre* mouse also exhibit “probable bone marrow failure” as evidenced by an increase in spleen weights found in the *Esr1/Kiss1-Cre* mouse?

As discussed above, no differences were noted in spleen weights in *Esr1^{Nkx2-1Cre}* females (Fig S3E), unlike the phenotype observed for *Esr1^{Kiss1Cre}* mutant female mice that is easily detected by 14 wks (Fig 6E).

4. Page 5, line 10 – change to “circulating catecholamines were not lower in the mutant.” If the data is not significant than the levels are not lower.

Motivated by this comment, we remeasured catecholamines levels in a new cohort of 7-8 wks old *Esr1^{Nkx2-1Cre}* females (n = 6). These new data shown in a revised Fig S2G, allow us to say with confidence that “circulating catecholamines were not lower in mutant females.”

5. Page 5, line 5 – How old are these OVX females? It would be helpful to the reader to state the ages for every experiment clearly as there are multiple ages used throughout leading to confusion.

As suggested by Rev1 and to avoid confusion we have revised our figure legends, text, and materials and methods to include the ages of all experimental animals.

6. 4.5-week-old females are peripubertal and should be described as such or at least juvenile throughout the text. I find juvenile used twice.

We now refer to 4.5 wk old mice as either peripubertal or juvenile throughout the text.

7. Due to the rather extreme density in bones, did the authors observe in differences in muscle strength or general mobility beyond the decrease in nighttime locomotor behavior in the KO (Fig. 1E)?

The idea that increased lean mass could be linked to a change in muscle is quite interesting and one that we wanted to explore in more depth. Using age (14 wks) and weight-matched female mice, we assessed their mobility (Rotarod) and overall muscle strength (Grip Strength). In both assays, neither mobility nor grip strength (average or maximal) were altered in *Esr1^{Nkx2-1Cre}* female mutants (Fig S2D, E).

8. In Figure 2E, samples sizes are states as 2 for *Esr1/fl* at 12 weeks and for *Esr1/Nkx2-1Cre* at 54 weeks. How can you then have a valid SEM error bar and statistical analysis with an ANOVA on a sample size of 2?

To increase the power of our statistics assessing volumetric bone mass over time the number of animals was increased for younger (12 wks) and older females (54-75 wks of age), as shown in Fig 2F). 2-Way ANOVA for genotype is ($F_{1, 50} = 172.1$, $P < 0.0001$), as stated in the legend.

9. A final point, POMC and KNDy neurons talk to each other (Fu & van den Pol, 2010, J Neurosci; Nestor et al 2016 Mol Endocrinol). One potential pathway is KNDy neurons modulate POMC tone and thus altering the hindbrain circuit upstream of bone remodeling all under the control of ER α activation in KNDy.

We also wonder if the elevated bone mass in mutant *Esr1^{Kiss1Cre}* females results from a change in other neuronal subpopulations, such as POMC, that directly communicate with KNDY expressing neurons and have included this possibility in our revised discussion. Such a scenario would be quite interesting, and might suggest a neuronal rather than a humoral based molecular mechanism for this bone phenotype.

Reviewer # 2

1. It is somewhat unclear if the central Kiss1 effect is ligand (E2) dependent or independent? Can the authors clarify this issue?

Rev2 raises an important question by asking if the high bone mass observed in *Esr1^{Kiss-Cre}* females depends on peripheral estrogen? Similar to what was observed in *Esr1^{Nkx2-1Cre}*, the high bone mass in *Esr1^{Kiss-Cre}* females appears to be uncoupled from changes in E2 or T (Fig S6E). An unequivocal answer to this question is challenging. Indeed, surgical removal of ovaries prior to development of the bone phenotype (preferably at 2 wks of age) would be confounded by the documented detrimental effects on post-natal skeletal development (Borjesson AE., et al., 2010 *JMBR*). Thus, it is formally possible that the bone phenotype in *Esr1^{Kiss-Cre}* females is partially dependent on a change in E2 levels as discussed in our revised text.

2. The remarkable sex difference is striking. Is this due to sex steroids or other factors? For example, is there an effect of Kiss1 or Nkx2 Cre deletion in male mice gonadectomized well before puberty (eg at 2-3 weeks of age)? Do the authors have an explanation for this marked sex difference?

We chose to directly test the provocative notion raised by Rev2 that circulating gonadal androgens in juvenile male mice might mask an elevated bone mass and thus explain the “remarkable” sex-difference. To do this, *Esr1^{fl/fl}* and mutant *Esr1^{Nkx2-1Cre}* males were surgically castrated at 3 wks of age. We failed to observe any changes in bone mass or skeletal microarchitecture between control or mutant males after depleting gonadal androgens (4 wks post-castration) (Fig S3D), thus eliminating this interesting hypothesis. In unpublished data with the Chan group (Stanford) we confirm that there are major changes to a stem cell niche in female, but not in male mutant bones. The most likely explanation(s) is that there are fundamental organizational and functional (ephys) sex-differences in ARC^{Kiss1} neurons as noted by others (Knoll JG., et al., 2013, *Front. Endo*, Yeo SH., et al., 2016, *J. Neuroendo*, de Croft et al., 2012, *Endo*). Clearly, future studies will be aimed at explaining how these sex-differences in bone arise.

3. A significant concern is that the mice used in the studies are in a very heterogeneous background, which can have significant effects on bone. Can the authors perform at least one of the key experiments (eg, AAV2 Cre) in a pure background (eg B6) to alleviate this concern?

As correctly noted by Rev2, genetic backgrounds can modestly affect bone mass. Recall that the mixed genetic background of *Esr1^{Nkx2-1Cre}* females contains CD-1;129P2. ER α KO^{ARC} are 129P2. *Esr1^{Kiss1Cre}* females are enriched for C57BL/6 because the Kiss-Cre KI line is on a pure C57BL/6 background. Documented %BV/TV for all strains used in our study range from 19% for C57BL/6 to

23% for 129SV (Sabsovich et al., 2008, *Bone*). Thus, we think it is highly unlikely that the significantly higher bone mass phenotypes observed in our three different mouse models reflect an inadvertent strain bias towards higher bone mass, especially given that the line containing the most C57BL/6 (*Esr1^{Kiss1Cre}*) has the highest bone mass change (460% increase). Notably, none of our models is on a C3H background, which exhibits the highest bone mass. At a practical level, the additional 8-10 months that would be required to generate/obtain the *Esr1^{fl/fl}* allele on a pure background, establish coordinates, and then carry out AAV2-Cre injections don't appear to be well-justified at this juncture.

4. The biological relevance of this central estrogen pathway for bone is still unclear. Perhaps the authors could provide a summary diagram showing how they believe this circuit regulates energy and bone metabolism. How important is this pathway when loss of estrogen consistently leads to low, and not high, bone mass? Clearly the peripheral actions of estrogen on bone are dominant. This should be acknowledged and discussed.

As noted by Rev2, our findings are consistent with the fact that loss of ovaries or gonadal estrogen with age has a major detrimental impact on bone mass. Thus, one would like to know when and how important this pathway is. As mentioned in our text, manipulation of other ARC neurons reveals that the hypothalamus plays an important role in restraining bone, albeit with much more subtle phenotypes compared to our findings. Similar to other genetic perturbations in mice (Leptin KO), we reveal an extreme phenotype. As discussed in our paper and is shown in a new schematic (Fig 6F), we speculate that this pathway in the ARC normally coordinates energetic allocation to restrain bone building and maximize fertility. Estrogen signaling in other brain regions would further coordinate reproduction (AVPV), energy expenditure (VMHvl) and sexual behavior (BNST, MEA). We would argue that knowing how these bones become so dense might counteract age-related bone loss that occurs in an estrogen-depleted state. We have modified our text to discuss the dominant role of peripheral estrogens on bone metabolism.

5. The authors are perhaps too dismissive of sympathetic outflow. They do find that the KO mice have a 50% reduction in NE (Fig. S2E) and that bAR2 is 50% lower (Table S1) in the KO mice. Lack of statistical significance here is likely just a matter of sample size. A more complete assessment of additional bAR target genes as described in the papers from the Karsenty lab should be performed.

We agree with Rev2 that we did not fully explore whether decreased sympathetic outflow might play a role in the high bone mass phenotype observed in female mutant mice. To address this and as mentioned above, we remeasured circulating catecholamines in a new cohort of 7-8-week-old age-matched wild type *Esr1^{fl/fl}* and mutant *Esr1^{Nkx2-1Cre}* females and find no differences (Fig S2G). Additionally, we also measured circulating levels of ACTH in these cohorts and find no differences (Fig S3C). We examined transcripts implicated by Karsenty and others that are associated with a change in sympathetic tone in our bone RNA-seq dataset, including *c-fos*, *IL-6*, *PGE2*, *AP-1*, *Tnfsf1*, *C-myc*, *Esp*, *Clock*, *Adrb2* and *Ucp1* (adj. p-values \geq 0.8). (Dataset S1), no differences are detected. Finally, in *ER α KO^{ARC}* females we find no changes in BAT gene signatures or morphology that would suggest a significant decrease in sympathetic outflow (Figure 1). Thus, there are no compelling data obtained thus far, which would suggest changes (Up or Down) in sympathetic outflow.

6. *Nkx2-1* is also expressed in the thyroid. Was ER α deleted in the thyroid and were thyroid hormone levels altered in the KO mice?

Please see comments for Rev1 above (#1).

7. I realize the authors are enthusiastic about their work, but they should refrain from using words like “astonishing” or “incredible”, etc to describe their findings. Please let the reader draw their own conclusions.

We have removed these single descriptors from our manuscript.

8. Fig. 2C – it is clear that the KO mice are not protected against ovx induced bone loss. It seems from Fig 2 that on a percentage basis, they lose as much bone or more than WT mice. These data should be shown (ie, percentage/absolute bone loss following ovx vs sham operated mice). Lack of protection from ovx bone loss would also argue that the peripheral effects of estrogen remain dominant, a point the authors tend to gloss over.

Rev2 correctly noted that mutant OVX *Esr1^{Nkx2-1Cre}* females lose bone. In fact, the percentage of bone loss in female mutants is higher than their wild type littermates (72% versus 43%), but they have a much higher starting point (Figure 2). This is consistent with others that report the highest OVX-mediated trabecular bone loss occurs in mice that have highest starting %BV/TV at baseline (Bouxsein et al., 2005 JBMR). As noted by Rev2, these data imply that the physiological effects of removing an ovary are dominant over the manipulations of estrogen signaling in the ARC, as our data on the ER α KO^{ARC} OVX females also show. Unfortunately, this surgical castration model in young female mice does far more than simply depleting estrogen – and in many ways is an imperfect model for both menopause and for assessing the effects of peripheral estrogen. To this point, we have now assessed older intact *Esr1^{Nkx2-1Cre}* female mutants at 1.5 years, which are no longer fertile and would have a >90% drop in E2 (Nilsson et al, 2015 Endo). Note, that neither LH nor FSH differ significantly in mutant versus control females at this age (LH: 0.7 \pm 0.03 vs 0.3 \pm 0.07 pg/ml; FSH: 5.6 \pm 3.3 ng/ml versus 7.0 \pm 2.5 ng/ml, n = 5, 5). We find that **bone mass is far better preserved in older intact mutant females versus younger OVX mutant** females. Importantly, age-related bone loss appears less severe in mutant versus control females (52% versus 75% loss from peak bone mass, Figure 2). Clearly, our findings are consistent with the fact that removing ovaries or depletion of gonadal estrogen with age has a major detrimental impact on bone mass. However, it is also true that mutant females (OVX and older intact) continue to exhibit higher bone mass than their control littermates. We would argue that knowing how these bones become so dense might be manipulated or exploited to counteract age-related bone loss that occurs in an estrogen-depleted state. We thank Rev2 for suggesting that we make this point clearer as to the dominant role of peripheral estrogens on bone metabolism.

Fig-2. Bone Loss in OVX and Aged Control and *Esr1^{Nkx2-1Cre}* Females. Representative images of control versus mutant femur following OVX at 18 wks of age and from older females at 54-71 wks of age (OLD) (left panel). Bar graph of data for different cohorts showing %BV/TV values for distal femur with percentages above each bar of peak bone mass determined in intact females at 12 wks of age.

9. Fig 4C – similar issue. Sham operated mice need to be included here to show the magnitude of the bone loss in each ovx group relative to sham.

It is unclear what is meant here, perhaps Rev2 is referring to Fig 4G. Adding Sham cohorts to the acute KO studies would not change the outcomes or conclusions, especially since others have carried out a comprehensive study on female mice of different genetic strains to assess how sham surgeries affect trabecular bone, micro architecture, BFR, and MAR compared with baseline. They report no statistical differences in femoral and vertebral bone parameters between baseline and Sham-

operated intact 129 females, and a small decrease only for femoral trabecular bone mass in C57BL/6 females (Iwaniec et al, 2009, JBMR). As such, we assert that repeating these time-consuming studies is not well-justified and does not directly address or control for the experimental question, which was to ask if one could build up bone after severe bone loss following OVX.

REVIEWERS' COMMENTS:

Reviewer #1 (Remarks to the Author):

The authors have sufficiently addressed all of my concerns and criticisms.

Reviewer #2 (Remarks to the Author):

The authors have done a nice job addressing my concerns. Overall, this is an important paper on the role of central ER α signaling and its effects on bone mass in female mice.